# Diagnostic Ability of Structural Transcranial Sonography in Patients with Alzheimer’s Disease

**DOI:** 10.3390/diagnostics10070471

**Published:** 2020-07-10

**Authors:** Daiva Rastenyte, Vaidas Matijosaitis, Ovidijus Laucius, Rymante Gleizniene, Simonas Jesmanas, Kristina Jureniene

**Affiliations:** 1Department of Neurology, Medical Academy, Lithuanian University of Health Sciences, A. Mickevičiaus Str. 9., LT-44307 Kaunas, Lithuania; Vaidas.Matijosaitis@lsmuni.lt (V.M.); Ovidijus.Laucius@lsmuni.lt (O.L.); 2Department of Radiology, Medical Academy, Lithuanian University of Health Sciences, A. Mickevičiaus Str. 9., LT-44307 Kaunas, Lithuania; Rymante.Gleizniene@lsmuni.lt (R.G.); s.jesmanas@gmail.com (S.J.); 3Institute of Cardiology, Medical Academy, Lithuanian University of Health Sciences, A. Mickevičiaus Str. 9., LT-44307 Kaunas, Lithuania; Kristina.Jureniene@lsmuni.lt

**Keywords:** transcranial sonography, magnetic resonance imaging, volumetry, medial temporal lobe, Alzheimer’s disease

## Abstract

The aim of this study was to assess the diagnostic ability of transcranial sonography (TCS) for the evaluation of the medial temporal lobe (MTL) in Alzheimer’s disease (AD). Standard neuropsychological evaluation, TCS and 1.5 T MRI were performed for 20 patients with AD and for 20 age- and sex-matched healthy controls in a prospective manner. Measurements of the size of the third ventricle and heights of the MTL (A) and the choroidal fissure (B) were performed twice on each side by two independent neurosonologists for all participants. On MRI, both conventional and volumetric analyses of the third ventricle and hippocampus were performed. Receiver operating characteristic (ROC) curves analyses were applied. Height of the MTL on TCS had sensitivities of 73.7% (right)/63.2%(left) and specificities of 65% (right)/65–70% (left) Area under a curve (AUC) 75.4–77.2% (right), 60.4–67.8% (left)) for AD. A/B ratio on TCS had sensitivities of 73.7% (right)/57.9% (left) and specificities of 70.0% (right)/55.0% (left) (AUC 73.3% (right), 60.4% (left)) by the experienced neurosonologist, and sensitivities of 78.9% (right and left) and specificities of 60.0% (right)/65.0% (left) (AUC 77.8–80.0%) by the inexperienced neurosonologist for AD. On MRI, linear measurement of the hippocampus and parahippocampal gyrus height had sensitivities of 84.2% (right)/89.5% (left) and specificities of 80.0% (right)/85% (left) (AUC 86.1–92.9%) for AD. Hippocampal volume had sensitivities of 70% (right and left) and specificities of 75% (right)/80% (left) (AUC 77.5–78%) for AD. Atrophy of the right MTL in AD could be detected on TCS with a good diagnostic ability, however MRI performed better on the left.

## 1. Introduction

Alzheimer’s disease (AD) and other forms of dementia comprise the largest pool of all neurodegenerative disorders in Europe and in the world [1]. Despite huge advances in discovering AD biomarkers, such as cerebrospinal fluid (CSF) amyloid β1-42, CSF tau, amyloid positron emission tomography (PET) imaging, fluorodeoxyglucose (^18^F-FDG) PET imaging, volumetric magnetic resonance (MR) imaging, [2,3], a reliable, safe and affordable diagnostic test for the diagnosis of dementia is yet undiscovered. Transcranial B-mode sonography (TCS) is a relatively novel non-invasive ultrasound method, capable of evaluating brain tissue echogenicity through an intact skull bone [4]. Using TCS, major parenchymal structures from the lower brainstem up to the parietal lobe, and the whole ventricular system may be visualized on standardized axial, coronal or semi-coronal imaging planes. Brain stem structures visualized by TCS include the substantia nigra, red nucleus, and midbrain nuclei raphe. Echogenicity of basal ganglia (thalamus, lenticular nucleus, head of caudate nucleus) and other deep brain structures (e.g., the middle temporal lobe, insula) may also be assessed [5,6]. To our knowledge the most of cortical structures cannot be visualized routinely by means of the TCS, although there is some evidence of temporal lobe cortex visualization [7,8,9]. Certain acknowledged limitations of TCS are associated with insufficient temporal acoustic bone windows in 5% to 20% of Caucasian patients [10]. Diagnostic precision might depend on the experience and skills of the neurosonographer and could substantially drop down due to inaccuracies when searching for the same scanning plane [6,11]. Nevertheless, TCS has been proven as a reliable diagnostic tool in the differential diagnosis of Parkinsonian syndromes [12,13,14], focal and global brain atrophy [15], and was also correlated with cognitive performance in multiple sclerosis [15,16,17] and Parkinson’s disease [18].

Recently, a structural ultrasound technique of the medial temporal lobe (MTL) assessment was proposed as a possible additive tool for the diagnosis of AD with a sensitivity of 83% and a specificity of 76%—comparable to the results published for brain MR imaging (MRI) [19]. The authors of the study, however, emphasized the need to validate the TCS measurements by means of brain MRI or postmortem examination [9].

Keeping in mind that MTL atrophy is considered one of the hallmarks in AD [20], we aimed to evaluate the MTLs of AD patients comparing to healthy subjects using the structural ultrasound technique proposed by Yilmaz et al. [19]. Obtained estimates were used to assess diagnostic ability of a structural TCS to differentiate patients with AD from healthy controls. We also aimed to validate the TCS measurements against measurements of conventional and volumetric brain MRI.

## 2. Material and Methods

This prospective study was performed within a framework of a project funded by the Research Council of Lithuania “Radiofrequency Ultrasound-based Brain Tissue Assessment Method for the Diagnostics of Early Neurodegeneration (NeuroRD)” Reg. No.: MIP-17-457. The study design and consent procedures were approved by the Ethics Committee for Biomedical Research at the Lithuanian University of Health Sciences (No BE-2-728, 19 December 2017), Kaunas, Lithuania. All participants gave their signed informed consent prior to inclusion in the study. All procedures were in accordance with the Declaration of Helsinki.

### 2.1. Patients and Control Group

Twenty AD patients were prospectively selected from both outpatient and inpatient units of the Department of Neurology at the Hospital of Lithuanian University of Health Sciences Kauno klinikos (Kaunas, Lithuania) from March, 2018 till March, 2019. Neurologists who invesigated the patients for a possible cognitive disorder performed initial selection of potential study subjects. Patients were included in the study if they: (1) were diagnosed with possible sporadic Alzheimer’s disease (AD) based on the NINCIDS-ADRDA criteria [21]; (2) provided written consent; (3) had satisfactory acoustic window properties on at least one side for TCS; (4) were eligible for MRI. Patients were excluded from the study if they: (1) had bilateral temporal acoustic bone insufficiency; (2) had a major somatic disease (decompensated heart failure, terminal renal or hepatic dysfunction, active cancer, had diagnosed hemodynamically significant intracranial/extracranial artery stenosis or thrombosis); (3) had a severe mental disorder (psychotic type, severe depression or the Geriatric Depression Scale (GDS) score >10); (4) were on continuous or prolonged use of medications affecting cognitive functions; (5) had history of head trauma; (6) had history of cerebrovascular accident; (7) had a prominent neurological deficit (severe visual disturbance, aphasia, severe paresis, ataxia, evident extrapyramidal signs) or other evident neurological disorders. Brain MRI was performed for all study patients but neither PET scan nor CSF biomarkers were investigated.

Twenty age and sex-matched control subjects were recruited from healthy relatives of the patients provided they did not have any major somatic illness as listed above, cognitive impairment or any abnormal findings in brain MRI, and were not under investigation or treatment for any neurodegenerative disease (Figure 1).

All participants were Caucasians and most of them were of Lithuanian origin.

All participants completed a questionnaire on general demographic information and risk factors. Education was evaluated by duration of formal education in years. Family history was considered positive if there was at least one known AD case among first- or second-degree relatives. Each patient with AD was consulted by a psychiatrist and a psychologist to rule out pseudodementia.

Both patients and control subjects underwent a standard neurological examination and a standard neuropsychological evaluation by means of a Mini-Mental State Examination (MMSE) test [22] and a shorter version of the Geriatric Depression Scale (GDS) [23]. Subjects with a GDS score of 10 or more were excluded from the study. AD patients were also tested with the Blessed dementia scale [24] to assess the severity of daily life impairment. According to the severity of cognitive dysfunction and daily life impairment, AD patients were categorized to mild, moderate and severe AD subgroups. Mild dementia was defined as a MMSE score of 20 to 24; moderate—of 11 to 19, and severe—of 10 or less [22].

### 2.2. Transcranial Sonography (TCS)

The application of TCS was based on the TCS quality standards [6] as well as on the methodology proposed by Yilmaz et al. [19].

All AD patients and healthy controls underwent TCS with a commercially available ultrasound system, Voluson 730 Expert BT08 (General Electric (GE) Healthcare, Zipf, Austria). The ultrasound system was equipped with an electronic sectored PA2-5P phased array transducer, which has a working range of 1.3 to 4.0 MHz, 128 piezoelectric elements, a modifiable 90 degree visual angle, and is also adapted for tissue harmonic imaging, when the frequency is 2.0 MHz. Greyscale mode (B-mode or Brightness) has 8-bit depth (i.e., 256 grey tones), a dynamic range of 180 dB and a maximum depth of 30 cm. The system comes installed with algorithms to reduce noise and artefacts.

TCS imaging was done using the PA2-5P/NEURO transducer mode. Scans were done in B-mode without any colour coding or Doppler mode. Amplitude gain was changed manually, which was done by observing the live TCS image. The brightest point on the screen was used as a reference point for amplitude gain. Regulator sliders of time gain compensation were used and placed in a semicircle with convexity to the right so that the region of interest, the hypoechogenic MTL and the surrounding hyperechogenic cerebrospinal fluid space could be seen the brightest.

Patients and healthy controls were supine in a darkened room, scans were performed by placing the transducer at the level of the eyebrows with the front part pointed upwards, i.e., to the side of face at the pre-auricular temporal bone area. Scans were performed in two standard planes: (a) the mesencephalic or the midbrain plane, and (b) the diencephalic plane or the third ventricle plane. Measurement of the height of the third ventricle in the segment neighboring brain stem was assessed.

Subsequently the same probe was placed perpendicular to the orbitomeatal line to visualize the ipsilateral hypoechogenic MTL in the coronal plane. MTL was defined as hypoechogenic structure surrounded by hyperechogenic cerebrospinal fluid. When the best image depicting the MTL and the surrounding area was identified, it was followed by the assignment of two linear measurements. Measurements of the height of the MTL (measurement A) together with height of the choroidal fissure (measurement B) were made using an approach proposed by Yilmaz et al. [12]. Height of MTL was measured in the middle of MTL as a horizontal line between the choroidal fissure (CF) and the bottom of the temporal lobe. Height of the CF was measured as the extension of a horizontal line in the midpoint of MTL (Figure 2). The MTL height/CF height ratio (A/B ratio) was calculated. The most suitable image for measurement was selected based on best visual structure differentiation and closely located brainstem was considered as good landmark. The MTL and CF were measured in cross-section close to the anterior or mid-brain stem.

All patients had measurements of the third ventricle diameter and the size of MTL performed twice on each side by two independent neurosonologists (one with 13 years of experience and the other with 2 years of experience). The penetration depth of TCS was usually 16.8 cm, while the zoom was 1.6 times to measure MTL, choroidal fissure and third ventricle dimensions. The normative threshold value of the third ventricle diameter calculated in our laboratory was <1.0 cm (mean + 2SD) [25].

### 2.3. MRI Acquisition

All MRI scans were obtained using a 1.5 T Siemens MAGNETOM Avanto (Erlangen, Germany) scanner within 2 to 4 weeks from the examination by TCS. The imaging protocol included axial T2W/TSE/2 mm (TR 4740 ms, TE 3.37 ms, TI 1100 ms, flip angle 120), T1W/mpr/p2/iso (TR 3000 ms, TE 89 ms), T2W/fl2d/hemo (TR800 ms, TE 26 ms, flip angle 20), coronal T2W/TSE (TR 5000 ms, TE 93 ms, flip angle 150), DW/ADC (TR 3000 ms, TE 89 ms), axial and coronal T2W/FLAIR (TR 9000 ms, TE 98 ms, TI 2500 ms, flip angle 150) and sagittal T2W/spc2d/iso (TR 3200 ms, TE 379 ms) sequences of the entire brain. No contrast media was injected. No hardware or software upgrades of the MRI scanner were done during the study period.

All sequences (T2W/FLAIR, T2W, T2W/fl2d/hemo, DW/ADC) were used to eliminate intra- and extra-axial lesions (tumors, vascular pathology, etc.).

Conventional linear measurements on MRI were performed by an experienced neuroradiologist. One linear measurement of the hippocampal head and parahippocampal gyrus was obtained from the coronal T2W images, planned in parallel to the dorsal part of the midbrain (the same angle as in TCS). The width of the third ventricle was measured at the dorsal part of the ventricle from the axial T1W images.

Volumetric analysis of T1W axial images was performed using the FreeSurfer v6.0 (Harvard, MA, USA, http://surfer.nmr.mgh.harvard.edu/) software package freely available online and run on a Linux CentOS 7 operating system. FreeSurfer performs volumetric segmentation of most macroscopic structures in the brain [26]. Technical details of how each step of the analysis works were described extensively in the previous literature. Briefly, these steps include motion correction, removal of non-brain tissue, segmentation, correction of topological defects, definition of grey/white matter and brain tissue/cerebrospinal fluid boundaries, registration to a spherical atlas and calculation of volumetric parameters [27,28,29,30,31,32]. In our analysis the basic fully automated pipeline was run to evaluate the volume of the third ventricle, while the hippocampal volume was evaluated using the additional hippocampal subfield segmentation module [33]. To correlate better with the ultrasound technique only the subfields belonging to the head and body category (CA1, CA2+3, CA4, parasubiculum, presubiculum, subiculum, molecular layer, granule cell and molecular layer of the dentate gyrus [GC-ML-DG], hippocampus-amygdala transition area [HATA] and fimbria) were kept and added together to produce a total hippocampal head and body volume.

### 2.4. Statistical Analyses

A database was created using Microsoft Office Excel 2007. For statistical analysis data was exported to the statistical package IBM SPSS version 25 (Armonk, NY, USA). Normality of quantitative data distribution was assessed using the Jarque Bera test. Descriptive and quantitative data are given as mean ± SD or median (Me) and interquartile range (IQR). Data of the two groups were compared using Student’s *t*-test or Mann–Whitney *U* test as appropriate. Logistic regression was performed to assess the predictive characteristics of the MTL measurements between AD group and healthy controls, with age, sex and ventricle measurements included as covariates. Receiver operating characteristics (ROC) curves were analysed in order to define a cut-off value for the highest sensitivity and specificity of TCS and MRI. Pairwise comparison of ROC curves was done using STATA/IC 10.0 for Windows. Correlation analysis was performed by Spearman’s rank correlation. For the assessment of intra/inter-rater reliability of the measurements, intraclass correlations (ICC) were evaluated in SPSS Using Single-Rating, Absolute-Agreement, 2-Way Random-Effects Model. A *p*-value of less than 0.05 was used as the criterion for statistical significance.

## 3. Results

A summary of demographic and clinical characteristics of AD patients and age- and sex-matched healthy controls is presented in Table 1. In the AD group, 9 patients had mild, 9 patients had moderate, and 2 patients had severe dementia. AD patients had a significantly lower MMSE score and were less educated although the latter difference was of a borderline significance. Groups were similar with regard to age and sex (Table 1).

On TCS, AD patients had a significantly wider third ventricle (*p* < 0.001) measured from either side of insonation by both neurosonologists (Table 2). The height of the MTL was smaller in AD patients as compared with healthy controls although the difference was significant just on the right side for both neurosonologists. The height of the CF was larger in the AD group, but the difference was significant just on the left side by the inexperienced neurosonologist. Compared with AD patients, healthy controls had a larger A/B ratio although the difference was statistically significant just on the right side for both neurosonologists, and also on the left side for the inexperienced neurosonologist.

The height of the MTL as well as the A/B ratio (on the right side for both neurosonologists, and on the left side for the inexperienced neurosonologist) correlated with the MMSE scores (*p*-values 0.0001–0.029).

The single measure intraclass correlation (ICC) for the height of the MTL of the experienced neurosonologist’s intrarater reliability was good (0.88; 95% CI 0.78–0.93) on the right side and moderate (0.62; 95% CI 0.39–0.78) on the left side. The single measure ICC for the height of the MTL of the inexperienced neurosonologist’s intrarater reliability was good on both sides (0.75; 95% CI 0.57–0.86 on the right side, and 0.79; 95% CI 0.64–0.88 on the left side). The ICC for the height of the MTL between the two neurosonologists was 0.67 (95% CI 0.46–0.81) on both sides which indicates moderate interrater reliability.

The single measure ICC for the A/B ratio of the experienced neurosonologist’s intrarater reliability was moderate both on the left (0.56; 95% CI 0.31–0.74) and on the right side (0.55; 95% CI 0.30–0.74). The single measure ICC for the A/B ratio of the inexperienced neurosonologist’s intrarater reliability was 0.75 (95% CI 0.57–0.86) on the left side and 0.71 (95% CI 0.51–0.83) on the right side. The ICC for the A/B ratio between the two neurosonologists was 0.70 (95% CI 0.49–0.83) on the left side and 0.50 (95% CI 0.23–0.70) on the right side.

In order to assess the predictive power of the MTL height (A) and of the MTL height to choroidal fissure height ratio (A/B) as measured by TCS for the likelihood that the subject has AD, eight logistic regression models were constructed accounting for age, sex, education and third ventricle width as covariates separately for each of the neurosonologists and for each side (right or left). Just the first measurement of the neurosonologist was taken into account.

All regression models were statistically significant except one (with the height of the left MTL for the experienced neurosonologist), had a good quality of fit and explained 39%–72% of the variance in group membership and correctly classified from 75% to 90% of cases. An increase in the width of the third ventricle was significantly associated with AD independently of age, sex and education, and regardless the side of insonation or experience of the neurosonologist (Table 3). The height of the MTL as well as the A/B ratio were also significantly associated with AD regardless of age, sex, education and third ventricle size on the right side but not on the left (Table 3).

On MRI, AD patients, compared with healthy controls, had a significantly wider third ventricle and a significantly lower height of the hippocampus and the parahippocampal gyrus on both sides (Table 2). Volumetric analysis showed that healthy control subjects had a significantly larger volume of the right hippocampus compared to the left (*p* < 0.001) (Table 2). This difference between the right and the left sides diminished in Alzheimer’s disease patients. Patients with Alzheimer’s disease had a significantly decreased volume of the hippocampus of both sides compared with healthy subjects.

Based on ROC curves, optimal cut-off values for the measurements of TCS and MRI were defined in order to classify a subject into one of two groups—AD or HC. Maximal width of the third ventricle >6.9 mm as measured from the right side by both neurosonologists separated the patients from the healthy subjects with a sensitivity of 84.2% and a specificity of 80.0%. The area under the curve (AUC) by both neurosonologists ranged from 86.1% (95% CI 73.8–98.3%) to 87.8% (95% CI 76.1–99.4%). Maximal width of the third ventricle >8.45 mm as measured on MRI separated the patients from healthy subjects with a lower sensitivity of 78.9% and an equal specificity of 80%, and had an AUC of 87.5% (95% CI 75.8–99.2%). In the pairwise comparison of ROC curves for the TCS by both neurosonologists and the MRI, there were no differences in the AUC (*p* = 0.74–0.94).

On the right side, maximal height of the MTL <13.2 mm as measured by the experienced neurosonologist separated the patients from healthy subjects with a sensitivity of 73.7% and a specificity of 65% with an AUC of 75.4% (95% CI 59.8–91.0%). For the inexperienced neurosologist the best cutoff value was 14.25 mm. At this value, both sensitivity and specificity corresponded to those of the experienced neurosonologist. The AUC was 77.2% (95% CI 62.4–92.0%). Maximal height of the hippocampus and parahippocampal gyrus <13.75 mm as measured on MRI separated the patients from healthy subjects both with a higher sensitivity of 84.2% and with a higher specificity of 80.0%. The AUC was 86.1% (95% CI 74.5–97.6%).

On the left side, the best cut-off value of the maximal height of the MTL was 13.2 mm for both neurosonologists. At this value, sensitivity was just 63.2% for both raters while specificity was 65% for the experienced neurosonologist and 70% for the inexperienced neurosonologist. The AUC for both neurosonologists ranged from 60.4% (95% CI 41.7–79.1) to 67.8% (95% CI 50.1–85.4%). Maximal height of the hippocampus and parahippocampal gyrus <13.35 mm as measured on MRI separated the patients from healthy subjects both with a much higher sensitivity of 89.5% and with a higher specificity of 85.0%. The AUC was 92.9% (95% CI 85.1–100.0%).

In the pairwise comparison of ROC curves for TCS on the right side of insonation by both neurosonologists and MRI, there were no statistically significant differences in the AUCs (*p* = 0.30–0.31). On the left side, statistically significant differences in the AUCs of the ROC curves for TCS and MRI were found (*p* = 0.001–0.007).

For the experienced neurosonologist, the ratio of A/B < 1.7 on the right side separated the patients from healthy subjects with a sensitivity of 73.7% and a specificity of 70.0% with an AUC of 73.3% (95% CI 57.5–89.1%). For the inexperienced neurosonologist, the ratio of A/B < 1.9 separated the patients from healthy subjects with a sensitivity of 78.9% and a specificity of 60.0% with an AUC of 80.0% (95% CI 66.4–93.6%).

On the left side the best cut-off of the A/B ratio was 1.9 for the experienced neurosonologist and 2.0 for the inexperienced neurosonologist. At these values, a sensitivity was 57.9% and a specificity was 55.0% for the experienced neurosonologists with an AUC of 60.4% (95% CI 42.3–78.5%). The corresponding values for the inexperienced neurosonologist were 78.9% and 65.0%, respectively, with an AUC of 77.8% (95% CI 63.0–92.5%).

Maximal volume of the right hippocampus <2654.0 mm^3^ separated patients from healthy controls with a sensitivity of 70% and a specificity of 75% while maximal volume of the left hippocampus <2568.8 mm^3^ separated patients from healthy controls with a sensitivity of 70% and a specificity of 80%. The corresponding AUCs were 77.5% (95% CI 62.4–92.6%) and 78.0% (95% CI 62.7–93.3%), respectively.

## 4. Discussion

In this study, the diagnostic ability of TCS in the evaluation of the MTL and the third ventricle was compared with conventional and volumetric MRI measurements in patients with Alzheimer’s disease and in healthy controls. MTL atrophy could be detected on TCS with a good diagnostic ability on the right side but not on the left. The TCS measurements of the third ventricle were found to be of very good diagnostic ability and comparable with MRI. On MRI, conventional linear measurement of the MTL when compared with automated volumetry of the hippocampus had similar specificity but higher sensitivity.

To our knowledge, this study is one of the first attempts to investigate the MTL parameters in patients with AD. Previous studies by Yilmaz et al. [9,19] aimed to determine the value of TCS in assessing the MTL and attempted to differentiate pathologic atrophy from normal changes. In their study published in 2016, the sensitivity and specificity of the ratio of the height of the MTL to the height of the choroidal fissure in differentiating AD patients from healthy subjects were 83% (right)/73% (left) and 76% (right)/72% (left), respectively, that is, better when compared to the present study [19]. Intra- and inter-rater reliability of these measurements were reported as good (ICC 0.79) and moderate (ICC 0.69), respectively. In the present study, inter-rater reliability for the MTL measurements was also moderate. Intra-rater reliability for the height of the MTL was good on the right side for both neurosonologists while on the left it was moderate for the experienced neurosonologist and good for the inexperienced neurosonologist. Moderate inter-rater reliability raises issues about qualification, standardization, and experience of the specialists performing TCS. Lower diagnostic accuracy of TCS demonstrated in our study may also be related to a different ultrasound system compared with the one used in the study by Yilmaz et al. [19]. Moderate inter-rater reliability for the experienced neurosonologist may also be related to the handedness issue which was raised by other researchers [19,34].

In our study, a remarkable lateralization was demonstrated as higher sensitivities and specificities were found for the right side measurements, and this is in accordance with the findings in the previous study by Yilmaz et al. [19]. Plane instability when performing sonography with the non-dominant hand could be one of the possible explanations for the lateralized findings.

Recently, Yilmaz et al., 2020 [9] showed an improvement in discriminating AD patients from controls compared to their previous study [19] due to changed measurement technique. In the coronal plane, the width of the temporal horn (TH) was measured instead of the height of the MTL. Measurements of the CF and MTL were added and the “MTL atrophy score in sonography” (MTA-S) was created. A mean MTA-S (on both sides) of 8.4 mm separated AD from controls with a sensitivity of 85% and a specificity of 83% [28]. In addition, measurements of both THs and the third ventricle were combined to form the “ventricle enlargement score in sonography” (VES-S). A VES-S score of 15 mm separated AD from controls with a sensitivity of 84% and with a specificity of 93% [28]. Additionally, both the intra-rater and inter-rater reliability values were substantial for the CF, TH and MTA-S on both sides. Besides, ability of MTA-S to separate MTL atrophy scale of MRI [20] scores 0–1 (no atrophy) from 2–4 with 100% specificity was shown [9]. 

In our study, the TCS measurements of the third ventricle from the right side of insonation were found to be of very good diagnostic ability in the differentiation of AD patients from healthy controls, and comparable with that of MRI. This is in line with the results of previous studies [15] and confirms the knowledge that widening of the third ventricle is a strong biomarker of brain atrophy but not specific as it can be found in various disorders of the brain [18,35].

In a review by Frisoni et al., diagnostic accuracy of imaging marker (measured by visual, manual, semiautomated, or automated segmentation/computation) in separating AD from healthy was highest for amyloid imaging (pooled sensitivity and specificity 88% and 85%, respectively) and progressively lower for ^18^F-FDG-PET (sensitivity 86%, specificity 84%), single-photon emission computed tomography (SPECT) (sensitivity 76%, specificity 84%), and MRI (sensitivity 75%, specificity 81%,) [36]. In our study, linear measurements of the MTL on MRI separated the AD patients from healthy subjects with a higher sensitivity of 89.5% and with a similar specificity of 85.0%. Furthermore, in our study these conventional linear measurements performed slightly better than automated volumetry of the hippocampus by having higher sensitivities, while volumetry itself had comparable sensitivities and specifities to MRI based assessments of MTL atrophy reported in the previously mentioned review [36]. This shows that reliable measurements can be performed by experienced neuroradiologists on routine MRI examinations which do not require time consuming volumetric analyses. On the other hand, neither routine MRI nor volumetric MRI is superior to PET imaging using ^18^F-FDG, ^11^C-labeled Pittsburg Compound B or beta-amyloid radiotracers (e.g., Florbetaben, Florbetapir, Flutemetanol) for the diagnosis of AD [36,37]. Nevertheless, a low specificity of beta-amyloid PET imaging in mixed populations of AD patients and patients with mild cognitive impairment should be noted [37].

Yilmaz et al. [19] have speculated that the underlying neurodegenerative process in healthy control subjects could play a role in the lower specificity of TCS, as atrophy of the hippocampal area may precede even the mild cognitive impairment stage [38]. In our study, healthy control subjects were matched by age, underwent a full neurological examination and brain MRI. We cannot exclude a possibility of ongoing insidious neurodegeneration in our control subjects but MRI did not reveal any pathological changes, and the AUCs for the MTL measurements by MRI demonstrated excellent diagnostic ability on both sides (92.9%). Data published by some other authors also show that ante mortem MRI measures of hippocampal volume are not specific for AD, but do correlate with AD severity as measured by Braak stage at autopsy [39,40].

It is worth noting that in the present study healthy control subjects had significantly larger volume of the right hippocampus compared with the left, while in patients with AD this difference was not evident. In several other studies the right hippocampus was also found to be significantly larger than the left in controls and those with a mild cognitive impairment, but not in AD patients [41,42]. In a meta-analysis by Shi et al. [34], a consistent left-less-than-right asymmetry pattern was found, but with different extent in control (effect size 0.39), mild cognitive impairment (effect size 0.56), and AD (effect size 0.30) groups.

Our study has several limitations. First of all, neither neurosonologists nor neuroradiologists were blinded to the status of the participants in the study. Second, our sample was quite small and this could influence the results, especially those related with the uncertainty with the left side measurements. Furthermore, it was impossible to produce more detailed analyses with respect to the severity of the disease and related conditions. Sample size was predefined by a main task of an entire original project, i.e., to develop a radiofrequency ultrasound-based brain tissue assessment method for the diagnostics of early neurodegeneration. Third, there is always a question regarding qualification and experience of the raters involved in the study. As was stated by Klöppel et al., the accuracy of MRI-based diagnostics of neurodegenerative disorders depends on the level of expertise of the involved radiologists [43]. The same is true and for neurosonologists performing TCS [6]. Although one of the neurosonologists was quite experienced, taking into account issues about handedness, we cannot exclude some imprecision in the measurements. Finally, one can question accuracy of AD diagnosis in our patients since AD diagnostics in our study was based on clinical criteria and MRI while neither PET nor CSF biomarkers were tested to support the diagnosis.

Atrophy of the right MTL in AD could be detected on TCS with a good diagnostic ability, however MRI showed better diagnostic precision on the left. The TCS measurements of the third ventricle were found to be very sensitive but non-specific diagnostic estimates of neurodegeneration comparable to MRI. Our results show that TCS MTL measurements used in a present study could reveal good diagnostic ability on the right side comparable to MRI measurements. This TCS methodology could be used to strengthen the clinical diagnosis of Alzheimer’s disease or be implemented in cases when MRI is either unavailable or contraindicated [9]. Besides TCS could be used to monitor degeneration process in time performing repeated testing. This study also supports a need for further research and search for more diagnostic ultrasound-based biomarkers or estimates to increase reliability of easily accessible and affordable diagnostic technique of neurodegeneration. Larger cohorts of AD patients and healthy controls need to be investigated in order to confirm clinical suitability for the measurement technique for routine assessment of MTL atrophy.

## Figures and Tables

**Figure 1 diagnostics-10-00471-f001:**
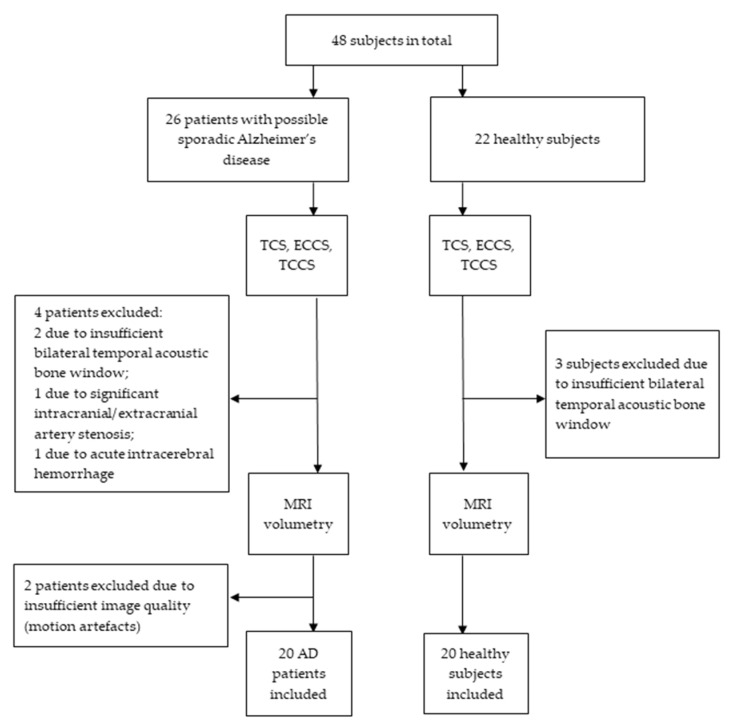
Study recruitment profile. TCS—transcranial sonography; TCCS—transcranial color-coded duplex sonography; ECCS—extracranial color coded duplex sonography.

**Figure 2 diagnostics-10-00471-f002:**
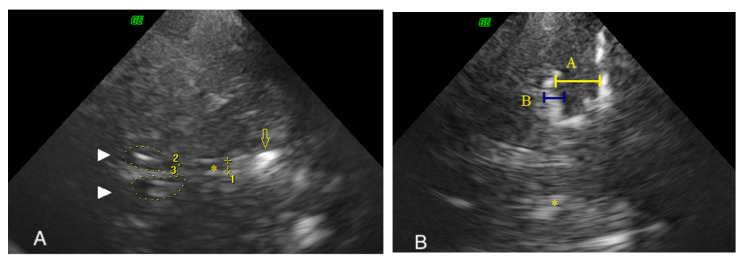
Ultrasound measurements of the third ventricle (**A**) and of the (**B**) medial temporal lobe on transcranial sonography. Brain structures visualized in axial plane (Figure 2A): arrowhead (2,3)—lateral ventricles; arrow—pineal gland; asterisk (1) —the third ventricle. Brain structures visualized in coronal plane (Figure 2B): A—height of the MTL, B—height of the choroidal fissure; asterisk—third ventricle.

**Table 1 diagnostics-10-00471-t001:** Clinicodemographic characteristics of patients with Alzheimer’s disease and of healthy controls.

Characteristics	AD, *n* = 20 Mean ± SD	HC, *n* = 20 Mean ± SD	*p*-Value
Age, years	71.8 ± 8.8	68.0 ± 6.5	n.s.
Sex, male % (n/20)	55 (11/20)	50 (10/20)	n.s.
Education, years	13.0 ± 3.6	15.2 ± 3.2	0.051
MMSE score	17.6 ± 5.4	29.2 ± 1.1	<0.001

AD—Alzheimers disease; HC—healthy controls; MMSE—Mini-Mental State Examination; SD—standard deviation; n.s.—not significan.

**Table 2 diagnostics-10-00471-t002:** Measurements of the third ventricle and the middle temporal lobe performed using transcranial sonography and magnetic resonance imaging with automated volumetric assessment software in patients with Alzheimer’s disease and in healthy controls.

Measurements	Left Side: Me (IQR)		*p*-Value	Right Side: Me (IQR)		*p*-Value
	AD	HC		AD	HC	
**TCS Measurements by the Experienced Neurosonologist**						
Width of the III ventricle, mm	8.2 (2.9)	5.6 (2.5)	<0.001	8.1 (2.5)	5.6 (2.0)	<0.001
Height of the MTL (A), mm	12.9 (2.4)	14.5 (2.9)	0.227	10.7 (4.8)	13.8 (3.2)	0.018
Height of the choroidal fissure (B), mm	7.0 (2.7)	6.5 (2.4)	0.718	7.3 (3.0)	7.3 (1.9)	0.758
A/B ratio	1.8 (0.7)	1.9 (0.6)	0.383	1.5 (0.7)	1.9 (0.5)	0.03
**TCS Measurements by the Inexperienced Neurosonologist**						
Width of the III ventricle, mm	8.7 (2.8)	6.2 (2.2)	<0.001	8.1 (2.5)	5.6 (2.0)	<0.001
Height of the MTL (A), mm	12.5 (3.0)	14.8 (2.6)	0.057	10.7 (4.8)	13.8 (3.6)	0.003
Height of the choroidal fissure (B), mm	7.5 (2.3)	6.7 (1.3)	0.044	7.5 (1.4)	7.0 (1.6)	0.079
A/B ratio	1.8 (0.4)	2.1 (0.6)	0.002	1.5 (0.7)	1.9 (0.7)	0.001
**MRI Measurements**						
Height of the hippocampus and parahippocampal gyrus, mm	12.4 (4.0) **	14.5 (3.3)	<0.001	13.0 (3.2)	15.2 (2.9)	<0.001
Volume of the hippocampus, mm^3^	2255.1 (697.5)	2656.7 ** (249.8)	0.002	2317.7 (705.5)	2801.8 (245.5)	0.002
	**AD**		**HC**		***p*-Value**	
Width of the III ventricle, mm	9.90 (3.0)		5.55 (3.6)		<0.001	

MTL—medial temporal lobe; TCS—transcranial sonography; MRI—magnetic resonance imaging; AD—Alzheimer’s disease; HC—healthy controls; Me—median; IQR—nterquartile range; ** *p* < 0.001 as compared with a right side of the same group.

**Table 3 diagnostics-10-00471-t003:** The predictive power of the medial temporal lobe height (A) and of the medial temporal lobe height to choroidal fissure height ratio (A/B) as measured by transcranial sonography for the likelihood that the subject has Alzheimer’s disease. Results of logistic regression analyses.

Parameter in Model	*β*	Odds Ratio	95% CI	*p*-Value
**Experienced Neurologist, Right**				
**1st Model**				
Width of the III ventricle, mm	1.103	3.013	1.319–6.885	0.009
Height of the MTL (A), mm	−0.47	0.625	0.417–0.936	0.022
**2nd Model**				
Width of the III ventricle, mm	0.948	2.58	1.237–5.381	0.012
A/B ratio	−3.196	0.041	0.002–0.917	0.044
**Experienced Neurologist, Left**				
**1st Model**				
Width of the III ventricle, mm	0.69	1.995	1.179–3.374	0.01
Height of the MTL (A), mm	−0.025	0.976	0.695–1.37	0.886
**2nd Model**				
Width of the III ventricle, mm	0.697	2.007	1.189–3.388	0.009
A/B ratio	0.347	1.414	0.205–9.776	0.044
**Inexperienced Neurologist, Right**				
**1st Model**				
Width of the III ventricle, mm	1.767	5.851	1.575–21.738	0.008
Height of the MTL (A), mm	−0.848	0.428	0.201–0.913	0.028
**2nd Model**				
III ventricle, mm	1.587	4.89	1.461–16.37	0.01
A/B ratio	−5.413	0.004	0.000–0.793	0.041
**Inexperienced Neurosonologists, Left**				
**1st Model**				
Width of the III ventricle, mm	1.158	3.184	1.497–6.773	0.003
Height of the MTL (A), mm	−0.063	0.939	0.671–1.314	0.7
**2nd Model**				
Width of the III ventricle, mm	1.228	3.416	1.451–8.04	0.005
A/B ratio	−2.685	0.068	0.003–1.637	0.098

95% CI—95% confidence interval.

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
