# Peer review of "Diagnostic Ability of Structural Transcranial Sonography in Patients with Alzheimer’s Disease"

_diagnostics, 2020, doi:10.3390/diagnostics10070471_

Round 1
Reviewer 1 Report
I reviewed with great interest this article entitled "Diagnostic ability of structural transcranial sonography in patients with Alzheimer’s disease" submitted by Daiva Rastenyte et al.
The objective of the study is relatively new and reflects the potential added value of transcranial sonography in patients with Alzheimer’s disease.
The prospective design study may provide important information despite the limited number of patients enrolled.
However, I have some request to the authors to improve the quality of the paper:
- The authors should define the exclusion criteria. In addition, a STARD FLOW showing included/excluded patients may help the reader
- the authors focused the evaluation on medial temporal lobe but AD may presents other morpho-functional changes in parietal lobe and retrosplenial cortex. Any results/evaluation on these areas?
- Did the authors compare the MRI and clinical signs with Brain-PET/CT 18F-FDG and/or 18F-amyloid radiopharmaceutical agents (eg.18F-Florbetaben) to confirm the diagnosis in vivo of AD? Any correlation with PET findings?
- Discussion could be improved discussing and comparing the performance of the most accurate diagnostic method (MRI and PET). A recent paper has been published in this regard and should be commented in the discussion. Yilmaz R, Granert O, Schäffer E, et al. Transcranial Sonography Findings in Alzheimer's Disease: A New Imaging Biomarker [published online ahead of print, 2020 Jun 3]. Befunde der transkraniellen Sonografie bei Morbus Alzheimer: Ein neuer bildgebender Biomarker [published online ahead of print, 2020 Jun 3]. Ultraschall Med. 2020;10.1055/a-1146-3036. doi:10.1055/a-1146-3036
Author Response
Replay to Review Report (Reviewer 1)
We thank to the Reviewer for his/her time, constructive critics and suggestions.
- The authors should define the exclusion criteria. In addition, a STARD FLOW showing included/excluded patients may help the reader.
We are thankful for this remark. We have edited exclusion criteria as follows (lines 80-88):
“Patients were excluded from the study if they: 1) had bilateral temporal acoustic bone insufficiency; 2) had a major somatic disease (decompensated heart failure, terminal renal or hepatic dysfunction, active cancer, had diagnosed hemodynamically significant intracranial/extracranial artery stenosis or thrombosis); 3) had a severe mental disorder (psychotic type, severe depression or the Geriatric Depression Scale (GDS) score >10); 4) were on continuous or prolonged use of medications affecting cognitive functions; 5) had history of head trauma; 6) had history of cerebrovascular accident; 7) had a prominent neurological deficit (severe visual disturbance, aphasia, severe paresis, ataxia, evident extrapyramidal signs) or other evident neurological disorders.”
Also, a flow chart was added (Figure 1. Study recruitment profile.) as it was suggested by the Reviewer (lines 93-97).
- The authors focused the evaluation on medial temporal lobe, but AD may present other morpho-functional changes in parietal lobe and retrosplenial cortex. Any results/evaluation on these areas?
We absolutely agree with a note of the Reviewer regarding morpho-functional changes not just in temporal lobe but also in other parts of brain. Nevertheless, evaluation of parietal lobe or retrosplenial cortex was out of scope of this study.
- Did the authors compare the MRI and clinical signs with Brain-PET/CT 18F-FDG and/or 18F-amyloid radiopharmaceutical agents (eg.18F-Florbetaben) to confirm the diagnosis in vivo of AD? Any correlation with PET findings?
We have not performed brain-PET/CT with 18F-FDG and/or 18F-amyloid radiopharmaceutical agents for our patients to confirm diagnosis of AD. We have added a sentence to subsection 2.1. to make this clear (lines 88-89): “Brain MRI was performed for all study patients but neither PET scan nor CSF biomarkers were investigated.” This was also mentioned in Discussion section as a limitation of our study (lines 392-394): “Finally, one can question accuracy of AD diagnosis in our patients since AD diagnostics in our study was based on clinical criteria and MRI while neither PET nor CSF biomarkers were tested to support the diagnosis.”
- Discussion could be improved discussing and comparing the performance of the most accurate diagnostic method (MRI and PET).
We thank the Reviewer for the suggestion. In response, we have expanded a comment on the topic (lines 348-353):
“In a review by Frisoni et al., diagnostic accuracy of imaging marker (measured by visual, manual, semiautomated, or automated segmentation/computation) in separating AD from healthy was highest for amyloid imaging (pooled sensitivity and specificity 88% and 85%, respectively) and progressively lower for 18F-FDG-PET (sensitivity 86%, specificity 84%), single-photon emission computed tomography (SPECT) (sensitivity 76%, specificity 84%), and MRI (sensitivity 75%, specificity 81%,) [36].”
Lines 360-364: “On the other hand, neither routine MRI nor volumetric MRI is superior to PET imaging using 18F-FDG, 11C-labeled Pittsburg Compound B or beta-amyloid radiotracers (e.g., Florbetaben, Florbetapir, Flutemetanol) for the diagnosis of AD [36,37]. Nevertheless, a low specificity of beta-amyloid PET imaging in mixed populations of AD patients and patients with mild cognitive impairment should be noted [37]”.
- A recent paper has been published in this regard and should be commented in the discussion. Yilmaz, R; Granert, O;Schäffer, E; Jensen-Kondering, U; Schulze, S; Bartsch, T; Berg, Transcranial Sonography Findings in Alzheimer's Disease: A New Imaging Biomarker]. Ultraschall Med. 2020; doi:10.1055/a-1146-3036.
We are especially thankful for this suggestion. Now a recent paper by Yilmaz et al. was commented in the discussion (lines 332-342):
“Recently, Yilmaz et al., 2020 [9] showed an improvement in discriminating AD patients from controls compared to their previous study [19] due to changed measurement technique. In the coronal plane, the width of the temporal horn (TH) was measured instead of the height of the MTL. Measurements of the CF and MTL were added and the “MTL atrophy score in sonography” (MTA-S) was created. A mean MTA-S (on both sides) of 8.4 mm separated AD from controls with a sensitivity of 85% and a specificity of 83% [28]. In addition, measurements of both THs and the third ventricle were combined to form the “ventricle enlargement score in sonography” (VES-S). A VES-S score of 15 mm separated AD from controls with a sensitivity of 84% and with a specificity of 93% [28]. Also both the intra-rater and inter-rater reliability values were substantial for the CF, TH and MTA-S on both sides. Besides, ability of MTA-S to separate MTL atrophy scale of MRI [20} scores 0-1 (no atrophy) from 2-4 with 100% specificity was shown [9].”
Reviewer 2 Report
The Authors assessed the possibility to evaluate the temporal lobe in patients with Alzheimer's disease by transcranial ultrasound.
Complete the background with proper sentences about parenchymal transcranial ultrasound. In particular, indicate the limits and the visible brain portions.
Re-write the aim in the last paragraph of the Introduction. It is too generic and not clear.
In Methods, the exclusion criteria are a repetition of inclusion criteria. In the current form, the inclusion/exclusion criteria are incomplete (for example, GDS < 10, head trauma, other neurological conditions, etc.).
Indicate A/B ratio in Methods.
Please, better indicate the aim of the use of ICC (I suppose for intra/inter-rater assessment).
The clinical application is not completely clear. Please, revise and add proper references.
Author Response
Replay to Review Report (Reviewer 2)
We thank to the Reviewer for his/her time, constructive critics and suggestions.
- Complete the background with proper sentences about parenchymal transcranial ultrasound. In particular, indicate the limits and the visible brain portions.
We thank the Reviewer for a suggestion. A text introducing parenchymal transcranial ultrasound was added to Introduction (lines 41-51):
“Using TCS, major parenchymal structures from the lower brainstem up to the parietal lobe, and the whole ventricular system may be visualized on standardized axial, coronal or semi-coronal imaging planes. Brain stem structures visualized by TCS include substantia nigra, red nucleus, and midbrain nuclei raphe. Echogenicity of basal ganglia (thalamus, lenticular nucleus, head of caudate nucleus) and other deep brain structures (e.g. the middle temporal lobe, insula) may also be assessed [5,6]. To our knowledge the most of cortical structures cannot be visualized routinely by means of the TCS, although there is some evidence of temporal lobe cortex visualization [7-9]. Certain acknowledged limitations of TCS are associated with insufficient temporal acoustic bone windows in 5% to 20% of Caucasian patients [10]. Diagnostic precision might depend on the experience and skills of the neurosonographer and could substantially drop down due to inaccuracies when searching for the same scanning plane [6,11].”
- Re-write the aim in the last paragraph of the Introduction. It is too generic and not clear.
We have rewritten the aim of the study as it was suggested by the Reviewer (lines 60-64):
“Keeping in mind that MTL atrophy is considered one of the hallmarks in AD [20], we aimed to evaluate the MTLs of AD patients comparing to healthy subjects using the structural ultrasound technique proposed by Yilmaz et al. [19]. Obtained estimates were used to assess diagnostic ability of a structural TCS to differentiate patients with AD from healthy controls. We also aimed to validate the TCS measurements against measurements of conventional and volumetric brain MRI.”
- In Methods, the exclusion criteria are a repetition of inclusion criteria. In the current form, the inclusion/exclusion criteria are incomplete (for example, GDS < 10, head trauma, other neurological conditions, etc.).
We are thankful for this remark. We are thankful for this remark. We have edited exclusion criteria as follows (lines 80-88):
“Patients were excluded from the study if they: 1) had bilateral temporal acoustic bone insufficiency; 2) had a major somatic disease (decompensated heart failure, terminal renal or hepatic dysfunction, active cancer, had diagnosed hemodynamically significant intracranial/extracranial artery stenosis or thrombosis); 3) had a severe mental disorder (psychotic type, severe depression or the Geriatric Depression Scale (GDS) score >10); 4) were on continuous or prolonged use of medications affecting cognitive functions; 5) had history of head trauma; 6) had history of cerebrovascular accident; 7) had a prominent neurological deficit (severe visual disturbance, aphasia, severe paresis, ataxia, evident extrapyramidal signs) or other evident neurological disorders.”
Also, a flow chart was added (Figure 1. Study recruitment profile.) as it was suggested by one the Reviewers (lines 93-97).
- Indicate A/B ratio in Methods.
A/B ratio was indicated in Methods as it was suggested (line 140): “The MTL height/CF height ratio (A/B ratio) was calculated.”
- Please, better indicate the aim of the use of ICC (I suppose for intra/inter-rater assessment).
The aim of the use of ICC was explained (lines 196-198): “For the assessment of intra/inter-rater reliability of the measurements, intraclass correlations (ICC) were evaluated in SPSS Using Single-Rating, Absolute-Agreement, 2-Way Random-Effects Model.”
- The clinical application is not completely clear. Please, revise and add proper references.
We have revised our comment on clinical application of the TCS as it was suggested (lines 398-406):
“Our results show that TCS MTL measurements used in a present study could reveal good diagnostic ability on the right side comparable to MRI measurements. This TCS methodology could be used to strengthen the clinical diagnosis of Alzheimer’s disease or be implemented in cases when MRI is either unavailable or contraindicated [9}. Besides TCS could be used to monitor degeneration process in time performing repeated testing. This study also supports a need for further research and search for more diagnostic ultrasound-based biomarkers or estimates to increase reliability of easily accessible and affordable diagnostic technique of neurodegeneration. Larger cohorts of AD patients and healthy controls need to be investigated in order to confirm clinical suitability for the measurement technique for routine assessment of MTL atrophy.”
Round 2
Reviewer 1 Report
The authors modified and improved the paper giving the clarification as requested by the reviewers.
The paper can be accepted in the present form.
Reviewer 2 Report
The manuscript has ben improved.